# The Occurrence and Meta-Analysis of Investigations on *Sarcocystis* Infection among Ruminants (Ruminantia) in Mainland China

**DOI:** 10.3390/ani13010149

**Published:** 2022-12-30

**Authors:** Zifu Zhu, Zhu Ying, Zixuan Feng, Qun Liu, Jing Liu

**Affiliations:** 1National Animal Protozoa Laboratory, College of Veterinary Medicine, China Agricultural University, Beijing 100193, China; 2Key Laboratory of Animal Epidemiology of the Ministry of Agriculture, College of Veterinary Medicine, China Agricultural University, Beijing 100193, China

**Keywords:** *Sarcocystis*, ruminants, mainland China, prevalence, meta-analysis

## Abstract

**Simple Summary:**

*Sarcocystis* can infect almost all warm-blooded animals, including humans. Ruminants (Ruminantia) are the intermediate hosts for *Sarcocystis*, representing a potential risk for public health. In China, there are many studies on the prevalence of *Sarcocystis*. However, the overall prevalence of *Sarcocystis* infection among ruminants in mainland China remains unclear and relevant risk factors need to be assessed to develop preventive measures. Therefore, we conducted this systematic review and meta-analysis to collect data on the prevalence and risk factors of *Sarcocystis* in ruminants in mainland China. Based on the results, we discussed the potential factors that may affect the prevalence of *Sarcocystis* in ruminants and the limitations in the current epidemiological investigation.

**Abstract:**

*Sarcocystis* is a zoonotic pathogen that threatens public health and the quality of food safety. To determine the *Sarcocystis* spp. prevalence in ruminants (Ruminantia) in China, we conducted a systematic review and meta-analysis. Data were collected from English databases (PubMed and Web of Science) and Chinese databases (Chinese Web of knowledge (CNKI), Database for Chinese Technical Periodicals (VIP) and Wan Fang databases). A total of 20,301 ruminants from 54 publications were evaluated. The pooled prevalence of *Sarcocystis* spp. among ruminants in mainland China was 65% (95% CI: 57–72%). Our results indicate that sarcocystosis is prevalent in ruminants, which show significant geographical differences. Therefore, it there is a need for continuous monitoring of infections of *Sarcocystis* spp. in ruminants to reduce the threat to human health and economic losses to the animal industry.

## 1. Introduction

*Sarcocystis* is a food-borne zoonotic pathogen that can infect humans and a variety of animals, there are 196 valid *Sarcocystis* species [1]. *Sarcocystis* has an obligate two-host life cycle, the intermediate hosts are usually herbivores, and the definitive hosts are usually carnivores or omnivores. *Sarcocystis* can form sarcocysts in the striated muscles and central nervous system of domestic animals such as sheep (*Ovis aries* Linnaeus, 1758), cattle (*Bos taurus* Linnaeus, 1758), and yaks (*Poephagus grunniens* Linnaeus, 1766) [1]. At present, a variety of *Sarcocystis* has been confirmed to have strong pathogenicity. Five known species have been described in cattle: *Sarcocystis cruzi* Hasselmann, 1926, *Sarcocystis hirsuta* Moulé, 1888, *Sarcocystis hominis* Railliet and Lucet, 1891, *Sarcocystis rommeli* Dubey et al., 2016, and *Sarcocystis heydorni* Dubey et al., 2015 [2,3]. Among them, humans are the definitive host of *S. hominis*, which can cause nausea, vomiting, and diarrhea [4]. *S. cruzi* is the most pathogenic species in cattle, and acute infection can lead to weakness, reduced milk yield, and economic losses in cattle farms [5]. Animals can be infected with *Sarcocystis* spp. by ingesting water or food contaminated with sporocysts or oocysts [1]. Humans become infected with *S. hominis*, *S. heydorni* and *Sarcocystis suihominis* tadros and Laarman, 1976 by eating undercooked meat containing mature sarcocysts.

Three validated species of *Sarcocystis* have been described in sheep in China: the pathogenic microscopic *Sarcocystis tenella* Railliet, 1886, *Sarcocystis arieticanis* Heydorn, 1985 and the non-pathogenic macroscopic *Sarcocystis gigantea* Railliet, 1886 [6]. Depending on the number and types of animals infected with *Sarcocystis*, it can cause anorexia, fever, muscle inflammation, abortion, premature birth, and even death [1]. Furthermore, the formation of sarcocysts in the muscles can affect the appearance and quality of meat, cause waste of meat products, and bring economic losses to animal husbandry. However, diagnosis is difficult due to the lack of specific clinical symptoms of *Sarcocystis*.

The animal industry is important in China, the annual production of meat from sheep in 2017 was 4851 thousand tons [7], and 66,728,000 tons of beef and 32,012,400 tons of milk in 2020 [8]. The economic significance of *Sarcocystis* in ruminants and its threat to human health via consumption of infected raw and undercooked meat (e.g., hot pots, roasts) highlights the importance of research on *Sarcocystis*. In addition, the relatively large number of surveys on the prevalence of *Sarcocystis* in China makes it necessary to conduct a systematic study to collect these data and to analyze the limitations and strengths of these studies. Meta-analysis and meta-regression are techniques that help to gain insight into the causes of such differences and to describe current knowledge in an evidence-based manner. Therefore, the aim of this study was to analyze the prevalence of *Sarcocystis* in ruminants and the factors that may influence the prevalence of *Sarcocystis* in ruminants in mainland China through meta-analysis and meta-regression.

## 2. Materials and Methods

### 2.1. Search Strategy and Inclusion Criteria

To evaluate the prevalence of *Sarcocystis* infection in ruminants, we performed a non-registered systematic review and meta-analysis of the literature published online. This system review and meta-analysis was conducted according to the recommendations of the Preferred Reporting Items for Systematic Reviews and Meta-Analyses (PRISMA) [9]. We performed the PRISMA 2020 checklist (Appendix A) to ensure the relevant information included was consistent with the study criteria. The meta-analysis was conducted by searching English databases (PubMed, Web of Science) and Chinese databases (Chinese Web of knowledge (CNKI), Database for Chinese Technical Periodicals (VIP), and Wan Fang databases) for publications related to *Sarcocystis* infection in ruminants in China, from 30 March 1983 to 1 January 2022. We searched databases using the following MeSH (Medical Subject Heading) terms alone or in combination: “*Sarcocystis*” or “*Sarcocystis* spp.” And “China” and “cattle” or “yak” or “sheep” or “goats” or “camelus” or “deer” or “ruminants” or “water buffaloes”.

Literature inclusion criteria: (1) the study population was limited to Chinese ruminants; (2) the types of studies were epidemiological and/or molecular studies of *Sarcocystis* spp. in ruminants; (3) the study must clearly report information on sources of samples, number of samples, infection prevalence of *Sarcocystis*, and detection methods for *Sarcocystis*; (4) the languages included in the study were limited to Chinese and English. 

Literature exclusion criteria: (1) the source of the reported samples and animal species was unclear, and the data were incorrect or incomplete; (2) samples were not randomly selected; (3) duplicate publications, conference publications, articles given to the editor, review articles, no full text or insufficient information, etc. 

### 2.2. Data Extraction 

Data were extracted independently by two investigators (Zhu Zifu and Feng Zixuan) and checked by other authors. From all included publications, we extracted the following information: first author, year of publication, year of sampling, geographical area of study, host, total number of samples examined, number of *Sarcocystis* spp. positive samples, diagnostic method, and age, gender, or identified species (if reported) (Table 1).

### 2.3. Statistical Analysis

This study was performed using R software for Meta-Analysis [63]. We used four methods to transform the original data to make them conform to the normal distribution: logarithmic transformation (PLN), logit transformation (PLOGIT), arcsine transformation (PAS), and double arcsine transformation (PFT). Statistical heterogeneity between studies was evaluated using the Cochran‘s Q statistics, *p*-value, and *I*^2^ statistics. The heterogeneity of included studies was used as the basis for selecting the effect model, and heterogeneity was considered insignificant when *p* > 0.1 and *I*^2^ < 50%, when *p* < 0.05 and *I*^2^ > 50%, the heterogeneity was considered significant. When the heterogeneity was not significant, a fixed-effect model was selected; otherwise, a random-effect model was used [64]. We conducted a sensitivity analysis in which one study was removed and the remaining studies were analyzed to assess the potential influence of the presence of outliers on each model per species. To account for sources of heterogeneity, we performed subgroup analyses based on region, host type, year of publication, and diagnostic method. The publication bias was evaluated by trim and fill analysis.

## 3. Results

### 3.1. Study Selection and Data Extraction

The search of five databases identified 332 records. After removing duplicate studies and preliminary screening, 244 papers remained. After screening titles and abstracts, 121 papers remained. We were unable to evaluate the full text of one article. After reading the full text, a total of 66 articles were excluded for the following reasons: review articles, non-ruminant animals, lack of *Sarcocystis* prevalence, and samples not randomly selected. Finally, a meta-analysis of 54 publications was performed, including 73 studies (Figure 1). 

The included studies were conducted in 9 provinces in China, including 4 provinces in northwest China, 2 provinces in southwest China, 2 provinces in central China, and 1 province in northeast China (Table 1, Appendix A). 

### 3.2. Prevalence of Sarcocystis in Ruminants in China

In this study, we used the arcsine transformation (PAS) conversion (the value of W was closer to 1, *p* > 0.05) for rate conversion (Table 2). High heterogeneity was shown in the included studies (*χ*^2^ = 8755.27, *d.f.* = 72, *p* < 0.001, *I*^2^ = 99.2%), the pooled infection rate of *Sarcocystis* in ruminants in mainland China was 65% (95% CI: 57–72%), as shown in Figure 2. A total of 20,301 ruminant samples were detected, of which 11,555 were positive for *Sarcocystis*. The detailed *Sarcocystis* prevalence in ruminants in different articles ranged from 7% to 100%. The presence of publication bias in the included articles could not be directly determined from the funnel plot (Appendix A). Begg’s and Egger’s tests showed that there was no publication bias in the included studies (*p* = 0.1514 > 0.05, Appendix A). Sensitivity analysis showed that no study had a significant effect on pooled prevalence (Appendix A).

### 3.3. Subgroup Analysis

The infection rate varied significantly by region in China (*p* < 0.01), with the highest prevalence of *Sarcocystis* infection in northwest China (10,205/15,919, 67%, 95% CI: 59–75%) and the lowest in northeast China (7/24, 29%, 95% CI: 13–51%) (Table 3, Figure 3). Significant differences in *Sarcocystis* infection were found among provinces (*χ*^2^ = 30.05, *d.f.* = 8, *p* < 0.01, *I*^2^ = 99%). The prevalence of *Sarcocystis* infection (7%) was lowest in Yunnan and highest (100%) in Tibet. 

The studies included a variety of animals, including camels (*n* = 1), cattle (*n* = 9), goats (*n* = 6), sheep (*n* = 34), water buffalo (*n* = 2), and yaks (*n* = 21) (Appendix A). There were significant differences in the prevalence of sarcocystosis among the different hosts (*χ*^2^ = 6.03, *d.f.* = 5, *p* < 0.01, *I*^2^ = 99%). Among them, the highest infection rate of *Sarcocystis* was found in goats (71%, 1788/3528, 95% CI: 40–93%) and the lowest in water buffalo (24%, 58/770, 95% CI: 0–72%). 

Our results showed the existence of significant differences (*χ*^2^ = 29.07, *d.f.* = 3, *p* < 0.01, *I*^2^ = 98%) in *Sarcocystis* infection rates across publication years (Appendix A). The pooled prevalence was significantly higher before 2005 (7771/10,588, 85%, 95% CI: 77–90%) than in 2005–2010 (1048/1998, 46%, 95% CI: 33–60%), 2011–2015 (746/1580, 50%, 95% CI: 30–70%) and after 2016 (1990/6135, 52%, 95% CI: 33–71%).

## 4. Discussion

This review describes the current knowledge on the prevalence of *Sarcocystis* among ruminants in mainland China in a systematic manner. All 54 publications included in this study were from naturally infected ruminants in China, including sheep, yaks, cattle, goats, water buffalos, and camels. The included studies were published from 1983 to 2021 and reported data on 20,301 ruminants. The pooled prevalence of *Sarcocystis* in ruminants was 65% (95% CI: 57–72%), which was lower than the overall prevalence of sarcocystosis in domestic ruminants in Iran (74.40%, 95% CI: 64.01–83.56%) [65]. Although pooled prevalence estimates from different animal species may be considered of limited value, it enables us to statistically define differences in the prevalence of *Sarcocystis* spp. and may provide information for taking appropriate measures to improve public health. 

In this meta-analysis, based on the *I*^2^ test, we observed a high degree of heterogeneity regarding the prevalence of Chinese ruminants among eligible studies. This may be related to detection method, age, region, year of sampling, year of publication, and host. 

At present, there is no standard serological diagnostic method for the detection of *Sarcocystis*, which is usually detected by muscle squashing microscopic observation, histological examination, HCl–pepsin digestion method, or molecular detection with specific primers [66]. Most of the studies in this review used morphological observation as the detection method, and the detection rate of this method is closely related to the sample type, sample size, and infection density of *Sarcocystis*, so the detection rate may be lower than that of the real infection. In addition, PCR identification was performed only on a small number of samples identified as positive by morphological observation, which could not truly reflect the infected species of *Sarcocystis* in all positive samples. 

The age of the animals is considered to be an important factor because the likelihood of animal exposure to parasites increases with age [67]. In this review, only two studies investigated the prevalence of *Sarcocystis* at different ages. One study reported a significantly higher prevalence of *Sarcocystis* infection in adult sheep (91.1%, 1912/2108) than in 5–8 months old lambs (68.2%, 101/148) [26]. Another study found that the infection rate (75%, 15/20) and intensity of infection (9.35 sarcocysts/cm^2^) were significantly higher in adult yaks than in young yaks (10%, 2/20) and (0.15 sarcocysts/ cm^2^), respectively, indicating an increase in infection rate with the age of the animals [32]. 

According to the subgroup analysis, there were significant differences in the infection rate of *Sarcocystis* among ruminants in different regions. The data showed that northwest China had the highest infection rate among the five regions in China, with the research mainly concentrated in Qinghai Province. There was only one study in northeast China, and the results of this study could not truly reflect the infection of *Sarcocystis* in this area. In addition, in this review, only 9 provinces have reported infection with *Sarcocystis* in ruminants, and many provinces have not yet reported it. 

The infection of ruminants with *Sarcocystis* is related to their living environment and feeding management. Strengthening the breeding management and reducing the exposure of ruminants to potential sources of *Sarcocystis* such as wild animals, cats and dogs can greatly reduce the probability of infection in ruminants [68]. According to our analysis, ruminants had the highest rate of *Sarcocystis* infection before 2005, which might be related to the continuous improvement of feeding management in recent years. However, in this review, there is no information on the feeding management of ruminants and the existence of definitive hosts. 

*Sarcocystis* is a food-borne pathogen which can infect a variety of animals and humans [1]. In this review, the overall prevalence of camels (*Camelus dromedarius and Camelus bactrianus*, Linnaeus, 1758), cattle, goats (*Capra hircus*, Linnaeus, 1758), sheep, water buffalo (*Bubalus bubalis*, Linnaeus, 1758), and yaks was generally higher, at 59%, 56%, 71%, 69%, 24%, and 64%, respectively. Among them, cattle are the intermediate hosts of *Sarcocystis* spp. Five species of *Sarcocystis* have been identified in beef, namely *S. cruzi*, *S. hirsuta*, *S. hominis*, *S. rommeli*, and *S. heydorni* [2,3]. Humans are the final hosts of *S. hominis* and *S. heydorni*, which can cause gastrointestinal distress in humans [4]. In addition to the possibility of zoonosis, there is evidence that these protozoa are related to bovine eosinophilic myositis (BEM), a specific inflammatory myopathy with gray-green lesions that causes carcass necrosis and considerable economic losses [69]. However, there are only nine studies on sarcocystosis in cattle and beef in this systematic review. 

This meta-analysis showed that sheep are the most common animal studied in the investigation of sarcocystosis in China, and 34 studies have reported *Sarcocystis* infection in sheep. The results showed that the pooled infection rate of *Sarcocystis* spp. was 69% (95% CI: 59–78%) in sheep, which is second only to goats among ruminants in China. Of these, only 3 articles identified infected *Sarcocystis* species in sheep, which are *S. gigantea*, *S. tenella,* and *S. arieticanis*. *Sarcocystis* in sheep is widely distributed in the world. The prevalence of *Sarcocystis* in sheep was reported to be 63.83% in Iran [65], 96.9% in Mongolia [70], 95.8% in Brazil [71], and 13.20% in Egypt [72]. 

In this systematic review, after sheep, yak became the most common target of sarcocystosis studies in China. Yak meat contains abundant trace elements and essential fatty acids beneficial to human body [73]. Yak meat products are welcomed by consumers because of their excellent quality. There are about 1.3 million yaks in China, accounting for 90% of the world’s yak population [73]. This may reveal why there are many studies on *Sarcocystis* among yaks in China. The results of this study showed that the prevalence of *Sarcocystis* spp. was 64% (95% CI: 50–78%), with significant differences in the infection rate of *Sarcocystis* in yaks in different studies (*p* < 0.01), which may be related to a variety of factors, such as age of animals, number of samples, and detection methods. 

In this systematic review, only 6, 2, and 1 studies were available for goats, water buffalo, and camels, respectively. Although only a few studies were available, it shows that *Sarcocystis* spp. infection occurs in these animals and deserves further attention. The pooled infection rate of *Sarcocystis* in goats was 71% (95% CI: 40–93%), which was highest among ruminants. The prevalence of *Sarcocystis* among goats in China was lower than that reported in Bahia, Brazil (91.6%) [71], and Iran (82.36%, 95% CI: 54.74–98.35%) [65]. Only one study identified infected species of *Sarcocystis* in goats, *S. capracanis* and *S. hircanis* [26]. The pooled infection of water buffalo was 24% (95% CI: 0–72%). The two studies on *Sarcocystis* of water buffalo were nearly 30 years apart, which showed high heterogeneity (*I*^2^ = 96%, *p* < 0.01) [18,43]. As camels live mainly in desert areas, they are mainly used as vehicles and are not a major source of meat products [74]. In this review, there is only one study on *Sarcocystis* of camel published in 1994, with an infection rate of 59% (95% CI: 51–65%) [14]. 

Our meta-analysis has several limitations: (1) the prevalence of *Sarcocystis* spp. in ruminants has not been reported in many regions of China; (2) the number of eligible studies is limited; (3) most of the studies were of medium or low quality, mainly because few risk factors (e.g., age, sex, and the presence of the final host) were available for analysis of *Sarcocystis* infection; (4) data on the ruminant living environment are insufficient and there may be different degrees of environmental pollution, which may explain the reason for the differences in the prevalence of *Sarcocystis* spp.; (5) the detection methods used were relatively simple, and only a few studies reported the infection of *Sarcocystis* species.

## 5. Conclusions

This meta-analysis determined the pooled infection rate of *Sarcocystis* in ruminants, indicating that *Sarcocystis* is widely present in many regions and animals in China. However, information on risk factors associated with *Sarcocystis* infection is very limited, and high-quality investigations are needed to further determine *Sarcocystis* infections in ruminants in China.

## Figures and Tables

**Figure 1 animals-13-00149-f001:**
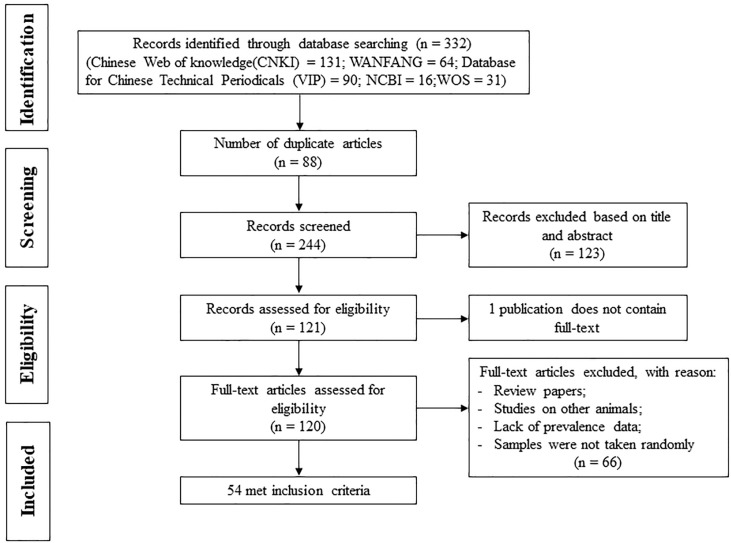
PRISMA flowchart of the systematic review and meta-analysis process shows the inclusion and exclusion of studies.

**Figure 2 animals-13-00149-f002:**
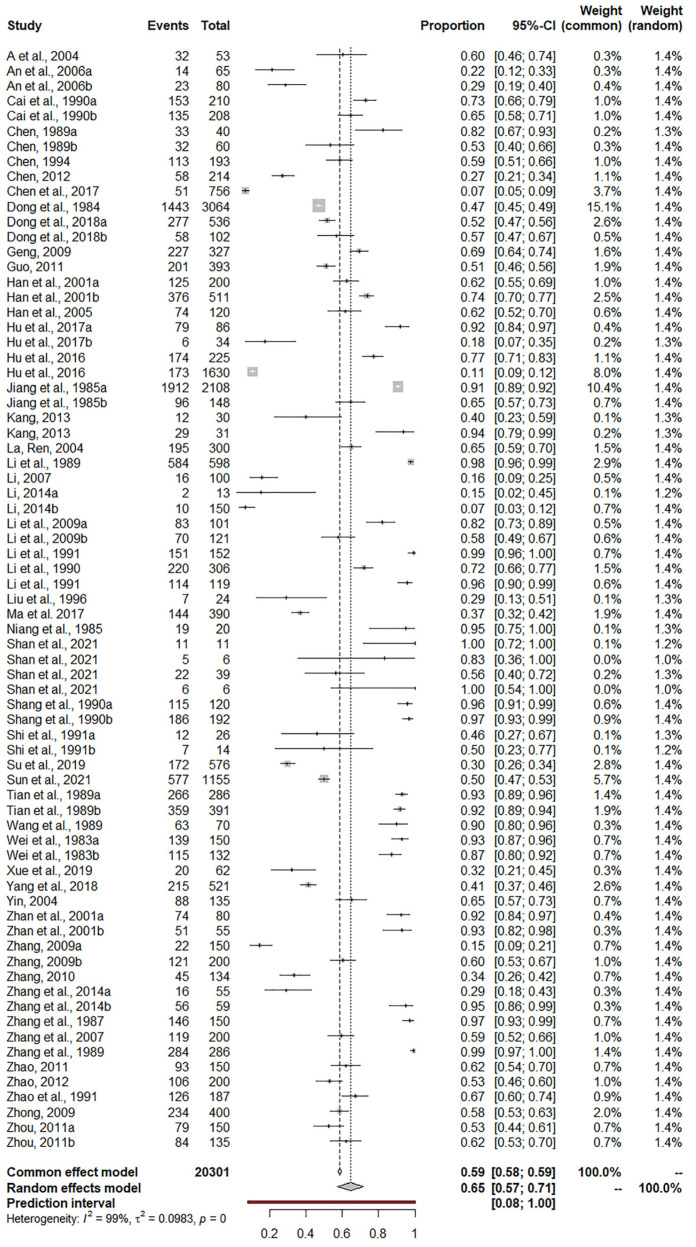
Forest plot of the prevalence of *Sarcocystis* infection among ruminants in China. CI: confidence interval. The first column “Study” included “Author + Year” [7,10,11,12,13,14,15,16,17,18,19,20,21,22,23,24,25,26,27,28,29,30,31,32,33,34,35,36,37,38,39,40,41,42,43,44,45,46,47,48,49,50,51,52,53,54,55,56,57,58,59,60,61,62].

**Figure 3 animals-13-00149-f003:**
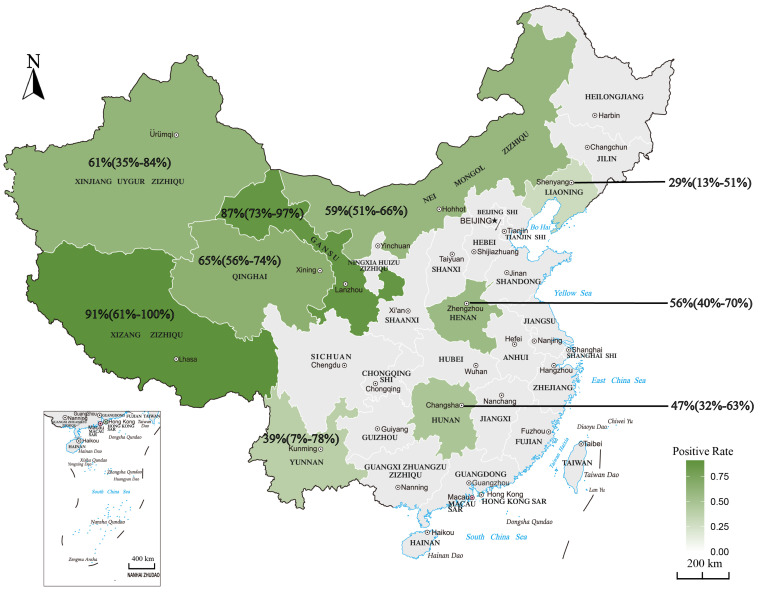
Prevalence of *Sarcocystis* infection among ruminants in mainland China.

**Table 1 animals-13-00149-t001:** Included studies of *Sarcocystis* infection in ruminants.

First Author, Year	Host	Province	Period of Study	Detection Method	Positive Samples/Total Samples (%)	Reference
A et al., 2004	Tibetan sheep	Qinghai	2003	Macroscopy, LM	32/53 (60)	[10]
An et al., 2006	Yaks	Xinjiang	2001	LM	14/65 (22)	[11]
An et al., 2006	Yaks	Xinjiang	2004	LM	23/80 (29)	[11]
Cai et al., 1990	Yaks	Qinghai	1988	Macroscopy, LM	153/210 (73)	[12]
Cai et al., 1990	Tibetan sheep	Qinghai	1988	Macroscopy, LM	135/208 (65)	[12]
Chen, 1989	Cattle	Henan	1985	Macroscopy, LM, H&E	33/40 (82)	[13]
Chen, 1989	Sheep	Henan	1985	Macroscopy, LM, H&E	32/60 (53)	[13]
Chen, 1994	Camel	Inner Mongolia	1992	Macroscopy, LM	113/193 (59)	[14]
Chen, 2012	Yaks	Qinghai	2011	LM	58/214 (27)	[15]
Chen et al., 2017	Water buffalo	Yunnan	n.s.	LM, TEM, PCR	51/756 (7)	[16]
Dong et al., 1984	Goats	Gansu	n.s.	LM; Pepsin digestion examination	1443/3064 (47)	[17]
Dong et al., 2018	Sheep	Henan	2014–2017	LM; Pepsin digestion examination; PCR	277/536 (52)	[7]
Dong et al., 2018	Sheep	Xinjiang	2014–2017	LM; Pepsin digestion examination; PCR	58/102 (57)	[7]
Geng, 2009	Yaks	Qinghai	2008	LM	227/327 (69)	[18]
Guo, 2011	Sheep	Qinghai	2011	Macroscopy, LM	201/393 (51)	[19]
Han et al., 2001	Yaks	Qinghai	n.s.	LM	125/200 (62)	[20]
Han et al., 2001	Sheep	Qinghai	n.s.	LM	376/511 (74)	[20]
Han et al., 2005	Sheep	Qinghai	n.s.	LM	74/120 (62)	[21]
Hu et al., 2017	Sheep	Yunnan	n.s.	LM, TEM, PCR	79/86 (92)	[22]
Hu et al., 2017	Cattle	Yunnan	n.s.	LM, TEM, PCR	6/34 (18)	[23]
Hu et al., 2016	Goats	Yunnan	2014–2015	LM, TEM, PCR	174/225 (77)	[24]
Hu et al., 2016	Cattle	Yunnan	2014–2015	LM, TEM, PCR	173/1630 (11)	[25]
Jiang et al., 1985a	Sheep	Xinjiang	n.s.	LM; Pepsin digestion examination	1912/2108 (91)	[26]
Jiang et al., 1985b	Sheep	Xinjiang	n.s.	LM; Pepsin digestion examination	96/148 (65)	[26]
Kang, 2013	Yaks	Qinghai	n.s.	LM	12/30 (40)	[27]
Kang, 2013	Sheep	Qinghai	n.s.	LM	29/31 (94)	[27]
La et al., 2014	Tibetan sheep	Qinghai	2003	LM	195/300 (65)	[28]
Li et al., 1989	Sheep	Qinghai	1986	LM	584/598 (98)	[29]
Li, 2007	Sheep	Qinghai	n.s.	Macroscopy, LM	16/100 (16)	[30]
Li, 2014	Goats	Qinghai	2011	Macroscopy, LM	2/13 (15)	[31]
Li, 2014	Sheep	Qinghai	2011	Macroscopy, LM	10/150 (7)	[31]
Li et al., 2009	Sheep	Qinghai	2006–2007	LM	83/101 (82)	[32]
Li et al., 2009	Yaks	Qinghai	2006–2007	LM	70/121 (58)	[32]
Li et al., 1991	Sheep	Qinghai	1989	Macroscopy, LM	151/152 (99)	[33]
Li et al., 1990	Cattle	Henan	n.s.	LM; Pepsin digestion examination, H&E	220/306 (72)	[34]
Li et al., 1991	Sheep	Qinghai	1990	Macroscopy, LM	114/119 (96)	[35]
Liu et al., 1996	Sheep	Liaoning	n.s.	Macroscopy, LM	7/24 (29)	[36]
Ma et al., 2017	Yaks	Qinghai	n.s.	LM	144/390 (37)	[37]
Niang et al., 1985	Yaks	Qinghai	1984	LM	19/20 (95)	[38]
Shan et al., 2021	Goats	Tibet	2019	Macroscopy, LM	11/11 (100)	[39]
Shan et al., 2021	Sheep	Tibet	2019	Macroscopy, LM	5/6 (83)	[39]
Shan et al., 2021	Yaks	Tibet	2019	Macroscopy, LM	22/39 (56)	[39]
Shan et al., 2021	Cattle	Tibet	2019	Macroscopy, LM	6/6 (100)	[39]
Shang et al., 1990	Yaks	Qinghai	1989	LM	115/120 (96)	[40]
Shang et al., 1990	Sheep	Qinghai	1989	LM	186/192 (97)	[40]
Shi et al., 1991	Cattle	Hunan	1990	Macroscopy, LM	12/26 (46)	[41]
Shi et al., 1991	Water buffalo	Hunan	1990	Macroscopy, LM	7/14 (50)	[41]
Su et al., 2019	Tibetan sheep	Qinghai	2017	Macroscopy, LM	172/576 (30)	[42]
Sun et al., 2021	Tibetan sheep	Qinghai	2017–2018	LM, H&E, TEM, PCR	577/1155 (50)	[43]
Tian et al., 1989	Yaks	Gansu	1985	Macroscopy, LM	266/286 93)	[44]
Tian et al., 1989	Sheep	Gansu	1985	Macroscopy, LM	359/391 (92)	[44]
Wang et al., 1989	Cattle	Xinjiang	1988	LM	63/70 (90)	[45]
Wei et al., 1983	Yaks	Gansu	1982	LM	139/150 (93)	[46]
Wei et al., 1983	Yaks	Gansu	1981	LM; Pepsin digestion examination	115/132 (87)	[47]
Xue et al., 2019	Cattle	Henan	2017	LM, H&E, PCR	20/62 (32)	[48]
Yang et al., 2018	Cattle	Henan	2014–2016	LM; Pepsin digestion examination, TEM, PCR	215/521 (41)	[49]
Yin, 2004	Sheep	Qinghai	2003	Macroscopy, LM	88/135 (65)	[50]
Zhan et al., 2001	Goats	Qinghai	2000	Macroscopy, LM	74/80 (92)	[51]
Zhan et al., 2001	Sheep	Qinghai	2000	Macroscopy, LM	51/55 (93)	[51]
Zhang, 2009	Yaks	Qinghai	n.s.	Macroscopy, LM	22/150 (15)	[52]
Zhang, 2009	Sheep	Qinghai	n.s.	Macroscopy, LM	121/200 (60)	[52]
Zhang, 2010	Yaks	Qinghai	2008	LM	45/134 (34)	[53]
Zhang et al., 2014	Yaks	Qinghai	n.s.	Macroscopy, LM	16/55 (29)	[54]
Zhang et al., 2014	Sheep	Qinghai	n.s.	Macroscopy, LM	56/59 (95)	[54]
Zhang et al., 1987	Yaks	Gansu	1986	Macroscopy, LM	146/150 (97)	[55]
Zhang et al., 2007	Sheep	Qinghai	2005–2006	LM	119/200 (59)	[56]
Zhang et al., 1989	Yaks	Qinghai	1986	H&E	284/286 (99)	[57]
Zhao, 2011	Sheep	Qinghai	2010	LM	93/150 (62)	[58]
Zhao, 2012	Tibetan sheep	Qinghai	2010	LM	106/200 (53)	[59]
Zhao et al., 1991	Yaks	Qinghai	1988	Macroscopy, LM	126/187 (67)	[60]
Zhong, 2009	Sheep	Qinghai	2007	LM	234/400 (58)	[61]
Zhou, 2011a	Sheep	Qinghai	2010	LM	79/150 (53)	[62]
Zhou, 2011b	Goats	Qinghai	2010	LM	84/135 (62)	[62]

LM: light microscopy; H&E: hematoxylin-eosin; PCR: polymerase chain reaction; TEM: transmission electron micrograph; n.s.: not specified.

**Table 2 animals-13-00149-t002:** Normal distribution test for the normal rate and the different conversion of the normal rate.

Conversion Form	W	*p*
PRAW	0.9351	0.001641
PLN	0.8341	1.38 × 10^−7^
PLOGIT	NaN	NA
PAS	0.96757	0.05696
PFT	0.96458	0.03809

PRAW: Original rate; PLN: logarithmic conversion; PLOGIT: logit transformation; PAS: arcsine transformation; PFT: double-arcsine transformation; NaN: meaningless number; NA: missing data.

**Table 3 animals-13-00149-t003:** Pooled prevalence of subgroup analysis (region, host and publish year) for *Sarcocystis* infection in ruminants in mainland China.

Subgroup Variable	No. Studies	No. Positive/ No. Tested	Prevalence (95% CI)	Heterogeneity		
				Q	*p*-Value	*I*^2^ (%)
Region						
Northeast China	1	7/24	29% (13–51%)	0.00	-	-
Central China	8	816/1565	54% (42–66%)	104.73	<0.01	93%
Northwest China	55	10,205/15,919	67% (59–75%)	4972.27	<0.01	99%
Southwest China	9	527/2793	64% (32–91%)	898.07	<0.01	99%
Host						
Camel	1	113/193	59% (51–66%)	0.00	-	-
Cattle	9	748/2695	56% (31–79%)	836.64	<0.01	99%
Goats	6	1788/3528	71% (40–93%)	209.62	<0.01	98%
Sheep	34	6707/9769	69% (59–78%)	2928.83	<0.01	99%
Water buffalo	2	58/770	24% (0–72%)	15.02	<0.01	93%
Yaks	21	2141/3346	64% (50–78%)	1473.82	<0.01	99%
Publish year						
<2005	33	7771/10,588	85% (77–90%)	2738.77	<0.01	98%
2005–2010	12	1048/1998	46% (33–60%)	337.88	<0.01	96%
2011–2015	12	746/1580	50% (30–70%)	352.37	<0.01	94%
>2016	16	1990/6135	52% (33–71%)	1551.60	<0.01	98%
Total	73	11,555/20,301	65% (57–72%)	8755.27	<0.01	99%

## Data Availability

Full data are available in the references list and Appendix A.

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
