# Peer review of "The Occurrence and Meta-Analysis of Investigations on Sarcocystis Infection among Ruminants (Ruminantia) in Mainland China"

_animals, 2022, doi:10.3390/ani13010149_

Round 1
Reviewer 1 Report
The manuscript concerns on the occurrence of the food-borne zoonotic apicomplexan Sarcocystic in ruminants in China. The data obtained are interesting, however, the paper needs some major revisions.
1. The title of the paper is inconsistent with the content of the manuscript. The paper does not contain information regarding the systematics of Sarcocystis in individual species of Ruminantia/Artiodactyla in China. After all, the results (Table 1) include data on only the genus Sarcocystis and not the species ? Without data on specific species, it is impossible to talk about systematics ! If the table will completed with data on Sarcocystis species everything (title) will be OK; although I doubt that such data exists.
2. Line 24: “Location” refers to habitat; “locality” to geographical region.
3. Keywords: why are there numbers next to words ?
4. Please provide full scientific names (with dates and author / authors) for all parasites and hosts, e.g. Sarcocystis accipitris Cern et Kvasnovsk, 1986.
Please check the notations of the species names, as there are errors, e.g. S. hirsuta - S. hirsute - the correct name is hirsuta.
5. “spp.” – should be in normal font.
6. For each common name, the scientific name must also be given, e.g. ruminants Ruminantia, even-toed ungulates Artiodactyla. Attention to the current systematics !.
Does the work (title: Systematic review and meta-analysis …among ruminants …) refer to ruminants Ruminantia, or the even-toed ungulates Artiodactyla ?
7. Line 66: please provide a definition of prevalence.
8. Lines 72-73: please provide whole names of Chinese databases; the abbreviation VIP can be misleading !!!
9. Lines 73: “from inception to January 1, 2021” – that is, since when, please specify?
10. Line 74: please provide the whole name for MeSH.
11 Lines 75-76: In Table 1 there is also another host species – water bufallo ?
12. Line 75: why "Sarcocystis sp." wasn't among the terms?
13. Lines 92-98: This paragraph is not clear; what the score 5 properly means, then 4, 3 and 2 ?
14. line 100-113: please provide appropriate references.
15. Table 1: The first (first author, year) and last columns (references) are the same. According to the Animals instructions for authors should be only citation/quote in the bracket.
Please provide legend: n.s., LM, H&E etc, etc; there are two entries H&E and HE.
16. Figure 2. First column – see comment 15 – should be [10] etc, etc.
17. Lines 152-153: “We assessed potential risk factors for heterogeneity by subgroup analysis, including region, host type, and year of publication” - this belongs to Material and methods.
18. I don't see supplementary materials - tables, figures !?
19. Figure 4: not clear - please change the font (bold, color) of the infection value.
20. Discussion: The description regarding individual hosts is incompatible. Sarcocystic species are not always given (yaks); if such data are missing then it should be clearly written. In addition, there are repetitions, e.g. line 261 and lines 272-273.
Author Response
Point 1. The title of the paper is inconsistent with the content of the manuscript. The paper does not contain information regarding the systematics of Sarcocystis in individual species of Ruminantia/Artiodactyla in China. After all, the results (Table 1) include data on only the genus Sarcocystis and not the species? Without data on specific species, it is impossible to talk about systematics! If the table will completed with data on Sarcocystis species everything (title) will be OK; although I doubt that such data exists.
Response 1: Thanks for your professional suggestions. The information regarding the systematics of Sarcocystis in individual species of Ruminantia/Artiodactyla in China is very limited. Although some of the included studies have identified the Sarcocystis species, the author only identified a small number of samples identified as positive by morphological observation, which could not truly reflect the infected species of Sarcocystis in all positive samples. In addition, most of the included studies did not identify Sarcocystis species. Therefore, in this study, we were unable to summarize the data on the Sarcocystis species.
As suggested, we have changed the title to “Systematic review and meta-analysis of investigations on Sarcocystis infection among ruminants (Ruminantia) in mainland China”. We look forward to your valuable suggestions, thank you!
Point 2. Line 24: “Location” refers to habitat; “locality” to geographical region.
Response 2: Thanks for your suggestion. We have modified it in our manuscript.
Point 3. Keywords: why are there numbers next to words ?
Response 3: We are sorry for our carelessness. This number is included in the manuscript template of this journal, and we forgot to delete it. We have modified it in our manuscript.
Point 4. Please provide full scientific names (with dates and author / authors) for all parasites and hosts, e.g. Sarcocystis accipitris Cern et Kvasnovsk, 1986.
Please check the notations of the species names, as there are errors, e.g. S. hirsuta - S. hirsute - the correct name is hirsuta.
Response 4: Thanks for your professional suggestions.
(1) Sarcocystis cruzi Hasselmann 1926
Sarcocystis hirsuta Moulé 1888
Sarcocystis hominis Railliet and Lucet 1891
Sarcocystis rommeli Dubey, Moré, van Wilpe, Calero-Bernal, Verma, Schares 2015
Sarcocystis heydorni Dubey, van Wilpe, Calero-Bernal, Verma, Fayer 2015
Sarcocystis suihominis tadros and Laarman 1976
Sarcocystis tenella Railliet 1886
Sarcocystis arieticanis Heydorn 1985
Sarcocystis gigantea Railliet 1886
cattle (Bos taurus)
yak (Poephagus grunniens)
sheep (Ovis aries)
goats (Capra hircus)
camels (Camelus dromedarius and Camelus bactrianus)
deer (Cervidae)
water buffalo (Bubalus bubalis)
ruminants (Ruminantia)
(2) We checked and revised all the notations of the species names in the manuscript.
Point 5. “spp.” – should be in normal font.
Response 5: We examined all the “spp.” in the manuscript and revised it.
Point 6. For each common name, the scientific name must also be given, e.g. ruminants Ruminantia, even-toed ungulates Artiodactyla. Attention to the current systematics !.
Does the work (title: Systematic review and meta-analysis …among ruminants …) refer to ruminants Ruminantia, or the even-toed ungulates Artiodactyla ?
Response 6: Thank you very much for your professional suggestions. The work refers to ruminants Ruminantia.
Point 7. Line 66: please provide a definition of prevalence.
Response 7: Prevalence refers to the proportion of certain disease cases among the total survey subjects in a specific time.
Point 8. Lines 72-73: please provide whole names of Chinese databases; the abbreviation VIP can be misleading !!!
Response 8: Thanks for your suggestions. We have modified it in our manuscript. CNKI(Chinese Web of knowledge), VIP(Database for Chinese Technical Periodicals).
Point 9. Lines 73: “from inception to January 1, 2021” – that is, since when, please specify?
Response 9: Thanks for your suggestion. From March 30, 1983 to January 1, 2022. We have modified it in our manuscript.
Point 10. Line 74: please provide the whole name for MeSH.
Response 10: MeSH (Medical Subject Heading), we have modified in our manuscript, Line 87.
Point 11. Lines 75-76: In Table 1 there is also another host species – water bufallo ?
Response 11: We are sorry for our carelessness, we forgot to put it in the manuscript. It has been revised in the manuscript.
Point 12. Line 75: why "Sarcocystis sp." wasn't among the terms?
Response 12: Thanks for your professional suggestions. We provided MeSH terms in the manuscript, but Sarcocystis sp. was included in the Entry terms. Because when Sarcocystis sp. is used as MeSH terms, the database retrieval result is 0. Therefore, “Sarcocystis sp.” wasn't among the MeSH terms.
Point 13. Lines 92-98: This paragraph is not clear; what the score 5 properly means, then 4, 3 and 2 ?
Response 13: We are sorry for our confusing description. The quality of the papers was assessed through a scoring approach, if a study matched one standard, it would be awarded 2 points, if the items in the study are unclear or not described, it would be awarded 1 point, and 0 point would be given if they do not match. Articles could be graded a total of 0-6. The score of 5 means that after scoring an article according to the scoring criteria, the score is 5 points.
Point 14. line 100-113: please provide appropriate references.
Response 14: Thanks for your professional suggestions. We have added the reference in our revised manuscript.
Point 15. Table 1: The first (first author, year) and last columns (references) are the same. According to the Animals instructions for authors should be only citation/quote in the bracket.
Please provide legend: n.s., LM, H&E etc, etc; there are two entries H&E and HE.
Response 15: Thanks for your suggestions. We have modified the format of the reference according to your suggestions. We think that readers can quickly capture some basic information according to the first column, such as the publication year of the article. In addition, there is a situation that an article comprises two studies, we add “a” and “b” after the author and the year to distinguish them, which may be more intuitive. Therefore, we would like to retain the contents of the first column. We look forward to your valuable suggestions.
We have provided the legend of the above abbreviations in the manuscript.
Point 16. Figure 2. First column – see comment 15 – should be [10] etc, etc.
Response 16: As mentioned in comment 15, we hope to retain the first column in the Figure 2, but do not know whether this is appropriate. We look forward to your further valuable comments, thank you.
Point 17. Lines 152-153: “We assessed potential risk factors for heterogeneity by subgroup analysis, including region, host type, and year of publication” - this belongs to Material and methods.
Response 17: Thanks for your suggestions. We have removed it from the results section.
Point 18. I don't see supplementary materials - tables, figures !?
Response 18: We uploaded the supplementary materials at the submission stage, you may need to download them.
Point 19. Figure 4: not clear - please change the font (bold, color) of the infection value.
Response 19: Thanks for your suggestions. We revised it in the manuscript.
Point 20. Discussion: The description regarding individual hosts is incompatible. Sarcocystic species are not always given (yaks); if such data are missing then it should be clearly written. In addition, there are repetitions, e.g. line 261 and lines 272-273.
Response 20: Thanks for your suggestions. According to your suggestions, we have adjusted and modified this part of the discussion (line 586-659).
After consideration, we have deleted line 272-273 in our manuscript.
Reviewer 2 Report
This study determined f investigations on Sarcocystis spp. infection among ruminants in mainland China, analyzed the research data through subgroup analysis and univariate regression analysis to reveal the factors leading to high prevalence. The workload is considerable, the assessment is very thorough, the methodology is largely correct, and I am pleased to see that more and more researchers in the veterinary field are using meta-analysis to study potential risk factors and disease prevalence. However, there are still some detailed issues that need to be further explored and modified.
1. Line 24, “P<0.01” Is italics required?
2. Please unify the reference format.
3. Line 74, I think the keywords obtained by using MeSH are more than the following.
4. Line 92, Why is the assessment conducted in this way?
5. Line96-98, I am doubt the description thoughtful, even though we can saw it in some other papers about high/medium/low quality description, personally I think it’s better not define other studies quality by your criteria.
6.Line 99, In addition to the univariate analysis, can the authors provide the results of the multivariate analysis, because I think the results of the multivariate analysis are also necessary.
7. Line 104, P value should be replaced by P-value.
8. Line 125, How were 72 studies obtained?
9. Line 145, Add a prediction interval to Figure 2.
10. Line 146, Modify the scale of Figure 3 to make it more harmonious in the article.
11. Line 170, Sarcocystis should be italicized.
12. Line 171, 72 articles have been selected from the above. Why is 73 articles in Table 3? Also, does I2 and P-value in Table 3 need italics?
13. Line 173, Why is the color of Hunan Province in Figure 4 different from that of provinces with the same positive rate? What is the reason?
14. Line208, There is only one study in Northeast China, so whether such analysis will have an impact on the grouping and produce bias.
15. Line 247, Is this conclusion supported by literature?
16. Line 285, Please provide relevant references.
17. Table 1: Please explain the abbreviations in the table
Author Response
Point 1. Line 24, “P<0.01” Is italics required?
Response 1: We are sorry for our negligence. We have revised it in our manuscript.
Point 2. Please unify the reference format.
Response 2: Thanks for your suggestions, we checked the reference format in the manuscript and made changes.
Point 3. Line 74, I think the keywords obtained by using MeSH are more than the following.
Response 3: We are sorry for our negligence. We have added it in our manuscript. We only wrote the MeSH terms used in this study in the manuscript. In fact, as you said, we obtained many Entry terms corresponding to the MeSH terms.
Point 4. Line 92, Why is the assessment conducted in this way?
Response 4: We originally intended to evaluate the quality of the included articles by setting a standard, but in the actual sampling process, the type and quantity of samples were random, so we realized that this scoring standard seemed unreasonable. Therefore, we deleted this part in the manuscript.
Point 5. Line96-98, I am doubt the description thoughtful, even though we can saw it in some other papers about high/medium/low quality description, personally I think it’s better not define other studies quality by your criteria.
Response 5: Thanks for your professional suggestions. As mentioned in comment 4, we have deleted this part in our manuscript.
Point 6. Line 99, In addition to the univariate analysis, can the authors provide the results of the multivariate analysis, because I think the results of the multivariate analysis are also necessary.
Response 6: Thanks for your professional suggestions. According to your suggestion, we have checked the relevant data and found that the interaction between factors may be ignored if only did the univariate analysis. However, multivariate analysis can combine multiple relevant factors from many studies. However, it is found that there are few factors in our study and some key information such as gender, age and diagnosis methods are lacking. Therefore, it is difficult to conduct multivariate analysis at present. In addition, we need to further study this part because we have little knowledge of multivariate analysis. We look forward to your further valuable comments, thank you.
Point 7. Line 104, P value should be replaced by P-value.
Response 7: Thanks for your professional suggestions. We have revised it in our manuscript.
Point 8. Line 125, How were 72 studies obtained?
Response 8: We are sorry for our mistakes, there are 73 studies in total, we have modified it in our manuscript. In this study, 54 articles comprising 73 studies. Since some published articles contain the results of more than one host infected with Sarcocystis, we have further subdivided these types of articles and summarized the data separately.
Point 9. Line 145, Add a prediction interval to Figure 2.
Response 9: Thanks for your professional suggestions. We have modified it in our manuscript.
Point 10. Line 146, Modify the scale of Figure 3 to make it more harmonious in the article.
Response 10: Thanks for your professional suggestions. We have revised it in our manuscript.
Point 11. Line 170, Sarcocystis should be italicized.
Response 11: Thanks for your suggestions. We have revised it in our manuscript.
Point 12. Line 171, 72 articles have been selected from the above. Why is 73 articles in Table 3? Also, does I2 and P-value in Table 3 need italics?
Response 12: We are sorry for our negligence. There are 73 articles in total, which we have modified in our manuscript. In addition, we have modified the font format in Table 3.
Point 13. Line 173, Why is the color of Hunan Province in Figure 4 different from that of provinces with the same positive rate? What is the reason?
Response 13: Since the infection rate of Sarcocystis infection in ruminants in Hunan Province is different from that in other provinces, we use different colors to mark this region.
Point 14. Line208, There is only one study in Northeast China, so whether such analysis will have an impact on the grouping and produce bias.
Response 14: Thanks for your professional suggestions. We performed a statistical test for publication bias, Begg's and Egger's tests showed that there was no publication bias in the included studies (P=0.1514>0.05).
Point 15. Line 247, Is this conclusion supported by literature?
Response 15: We are sorry for our mistakes. We know that the local people have the habit of eating raw yak meat, but we have not found relevant literature to support this conclusion. Therefore, we deleted this sentence from the manuscript.
Point 16. Line 285, Please provide relevant references.
Response 16: Thanks for your suggestion, we provided relevant references in our manuscript.
Point 17. Table 1: Please explain the abbreviations in the table.
Response 17: Thanks for your professional suggestions. We have revised it in our manuscript.
Reviewer 3 Report
The meta-analysis of sarcoystosis in ruminants in China seems to be well conducted. It doesn't seem to add much meaniful information.
All of the figures and tables need better legends.
Table 1. The abbreviations in detection methond need to be defined. Including percentage in parenthesis (%) in the positive/total column would be helpful.
Figure 1. 1 publications do
Figure 2 needs more description in the legend and in the text associated with the figure.
Similarly, more description of the significance of the funnel plot is needed in the text and the legend (Fig 3) needs to be expanded. Is this figure needed?
The significance of Table 2 is unclear. What are W and P? Is this table needed?
Table 3. What is the Total referrring to? And why is it 73 studies instead of 54?
Lines 16-17. There is no discussion of risk factors.
Lines 23-24. The sentence spanning lines 23-24 can be deleted. It is mentioned in the next sentence.
Line 24. sarcocytosis is not italicized.
Line 27. animal husbandry is probably better than breeding industry.
Author Response
Point 1. All of the figures and tables need better legends.
Response 1: Thanks for your suggestions. We checked the figures and tables in the manuscript and revised the legend.
Point 2. Table 1. The abbreviations in detection methond need to be defined. Including percentage in parenthesis (%) in the positive/total column would be helpful.
Response 2: Thanks for your professional suggestions. As suggested, we have revised it in our manuscript.
Point 3. Figure 1. 1 publications do
Response 3: Thanks for your suggestion, we have revised in our manuscript.
Point 4. Figure 2 needs more description in the legend and in the text associated with the figure.
Similarly, more description of the significance of the funnel plot is needed in the text and the legend (Fig 3) needs to be expanded. Is this figure needed?
Response 4: Thanks for your suggestion. We have revised the corresponding part of the manuscript.
After consideration, it seems that Figure 3 is not necessary in the manuscript, so we put it in the supplementary materials.
Point 5. The significance of Table 2 is unclear. What are W and P? Is this table needed?
Response 5: The W test, which stands for Shapiro-Wilk test, is an algorithm based on correlation. A correlation coefficient can be calculated, and the closer it is to 1, the better the data fit the normal distribution.
P stands for P-value. The W value is close to 1, and the P-value is much larger than 0.05, indicating that it conforms to the normal distribution.
Table 2 is the result of the transformation of the original data using four methods, which led us to choose the appropriate method. Therefore, we feel it is necessary to retain this table. Thank you!
Point 6. Table 3. What is the Total referrring to? And why is it 73 studies instead of 54?
Response 6: We are sorry for our confusing description. Total refers to a generalization of included studies.
In this study, 54 articles comprising 73 studies. Since some published articles contain the results of more than one host infected with Sarcocystis, we have further subdivided these types of articles and summarized the data separately.
Point 7. Lines 16-17. There is no discussion of risk factors.
Response 7: Thanks for your suggestions. After consideration, we have changed “risk factors” into “potential factors”. In the discussion section, we analyzed the potential factors that may affect the prevalence of Sarcocystis in ruminants, such as detection method, age, sex, host, etc.
Point 8. Lines 23-24. The sentence spanning lines 23-24 can be deleted. It is mentioned in the next sentence.
Response 8: Thanks for your suggestion. As suggested, we have deleted this sentence.
Point 9. Line 24. sarcocytosis is not italicized.
Response 9: Thank you, we have modified it.
Point 10. Line 27. animal husbandry is probably better than breeding industry.
Response 10: Thanks for your suggestions, we have modified it in our manuscript.
Round 2
Reviewer 1 Report
Comments:
1. Title.
Unfortunately, the authors of the manuscript did not comply with my comment; they themselves also admitted that they could not supplement the data with species from Sarcocystic genus, because this type of data is not found in the literature.
Systematic of organisms is the science that deals with classifying, cataloging and describing organisms based on the study of their diversity, origin and relatedness.
Without knowledge of specific species, it is difficult to talk about systematic here.
The title is inaccurate, misleading; after the name "systematic" the reader (scientist) expects something else, so the title should be changed to, for example, The occurrence and meta-analysis ….”.
2.
Unfortunately, the authors did not complete the manuscript with names and dates, they provided (with errors !!!) them only in the letter / answers to the reviewer ?
When first time a species is mentioned in the text, its whole name should be given, that is, the common (English) name, the scientific name with the author(s) and the date.
I would like to point out that such notations are regulated by the International Code of Zoological Nomenclature !!!, e.g.
Sarcocystis hirsuta Moulé, 1888 - always put a comma between the author(s) and the date.
Sarcocystis hominis (Railliet and Lucet, 1891) - the authors of the species are not always in brackets; authors are in the brackets if the species has previously been described under another name, e.g., another generic name !!!
Sarcocystis rommeli Dubey, Moré, van Wilpe, Calero-Bernal, Verma and Schares, 2015 - "et" or "and" is placed between the penultimate and last authors.
Please check the accuracy of the notations for the other/all species.
3. Unfortunately, the authors did not enter the whole, correct names of the hosts - they only gave the authors ?; for example, the correct name of the sheep is: sheep Ovis aries Linnaeus, 1758.
Author Response
Point 1. Title.
Unfortunately, the authors of the manuscript did not comply with my comment; they themselves also admitted that they could not supplement the data with species from Sarcocystic genus, because this type of data is not found in the literature.
Systematic of organisms is the science that deals with classifying, cataloging and describing organisms based on the study of their diversity, origin and relatedness.
Without knowledge of specific species, it is difficult to talk about systematic here.
The title is inaccurate, misleading; after the name "systematic" the reader (scientist) expects something else, so the title should be changed to, for example, The occurrence and meta-analysis ….”.
Response 1: Thanks for your professional suggestions, we have changed the title into “The occurrence and meta-analysis of investigations on Sarcocystis infection among ruminants (Ruminantia) in mainland China” in our manuscript.
Point 2.
Unfortunately, the authors did not complete the manuscript with names and dates, they provided (with errors !!!) them only in the letter / answers to the reviewer ?
When first time a species is mentioned in the text, its whole name should be given, that is, the common (English) name, the scientific name with the author(s) and the date.
I would like to point out that such notations are regulated by the International Code of Zoological Nomenclature !!!, e.g.
Sarcocystis hirsuta Moulé, 1888 - always put a comma between the author(s) and the date.
Sarcocystis hominis (Railliet and Lucet, 1891) - the authors of the species are not always in brackets; authors are in the brackets if the species has previously been described under another name, e.g., another generic name !!!
Sarcocystis rommeli Dubey, Moré, van Wilpe, Calero-Bernal, Verma and Schares, 2015 - "et" or "and" is placed between the penultimate and last authors.
Please check the accuracy of the notations for the other/all species.
Response 2: We are sorry for our mistakes. Thanks for your professional suggestions. We have revised the manuscript in the following format:
Line 40-42: “Sarcocystis cruzi Hasselmann, 1926, Sarcocystis hirsuta Moulé, 1888, Sarcocystis hominis Railliet and Lucet, 1891, Sarcocystis rommeli Dubey et al., 2016, Sarcocystis heydorni Dubey et al., 2015.”
Line 55: “Sarcocystis suihominis tadros and Laarman, 1976”
Line 58-59: “Sarcocystis tenella Railliet, 1886, Sarcocystis arieticanis Heydorn, 1985, Sarcocystis gigantea Railliet, 1886”.
Point 3. Unfortunately, the authors did not enter the whole, correct names of the hosts - they only gave the authors ?; for example, the correct name of the sheep is: sheep Ovis aries Linnaeus, 1758.
Response 3: Thanks for your professional suggestions. We have revised the manuscript in the following format:
Line 37-38: “…. sheep (Ovis aries Linnaeus, 1758), cattle (Bos taurus Linnaeus, 1758), and yaks (Poephagus grunniens Linnaeus, 1766)”.
Line 246-248: “In this review, the overall prevalence of camels (Camelus dromedarius and Camelus bactrianus, Linnaeus, 1758), cattle, goats (Capra hircus, Linnaeus, 1758), sheep, water buffalo (Bubalus bubalis, Linnaeus, 1758) and yaks….”.